# CTNet: A CNN-Transformer Hybrid Network for 6D Object Pose Estimation

## Abstract

Recent advances in 6D pose estimation primarily rely on CNNs, but they struggle to grasp long-range dependencies and the global context, which are essential for precise pose determination. Although deeper or expanded networks are commonly used to tackle this, they lead to significant computational burdens without fully addressing these constraints. To overcome these challenges, we present CTNet, a hybrid network that fuses the strengths of CNN and Transformer, aiming for accurate 6D pose estimation from a solitary RGB-D image. CTNet employs Transformer to capture elusive long-range dependencies and the global context, while lightweight CNNs adeptly extract detailed local features. This complementary approach offers a comprehensive feature representation, eliminating the necessity for excessively deep networks. To further bolster the CNNs' efficiency, we introduce the Hierarchical Feature Extractor (HFE), which enhances the C2f and ELAN modules for optimal feature extraction. Additionally, we integrate a CNN-based PointNet module, designed to extract vital spatial data from the point cloud. The Transformer element captures global contextual insights, which are then seamlessly integrated with the local and spatial features extracted by the CNNs to ensure precise 6D pose estimation. Experiments demonstrate that CTNet achieves high accuracy with nearly half the FLOPs of current methods on the LineMOD and YCB-Video datasets. Furthermore, the HFE is highly adaptable, showing excellent transferability across other 6D pose estimation architectures.

## 1 Introduction

6D pose estimation aims to determine the position and orientation of objects in 3D space (Xiang et al., 2014), with significant applications in complex robotic manipulation tasks (Tremblay et al., 2018; Collet et al., 2011), immersive augmented reality experiences (Marchand et al., 2015), and advanced autonomous driving systems (Chen et al., 2017; Geiger et al., 2012). In order to fulfill the real-time demands of these applications, pose estimation often needs to be processed on mobile computing platforms (Yang et al., 2024). However, the high computational complexity of current models poses challenges for efficient performance within such resource-constrained environments.

To tackle 6D pose estimation, researchers explore diverse approaches, leveraging CNNs and PCNs (Zhang & Liu, 2023; Yuan et al., 2018). Methods relying on texture information from RGB images (Peng et al., 2019; Tekin et al., 2018b; Kehl et al., 2017; Kendall et al., 2015) encounter difficulties such as decreased accuracy with weakly textured images and sensitivity to lighting variations. Alternatively, methods using geometric information from point clouds (Qi et al., 2017a;b) struggle with high redundancy, unstructured data, and sensitivity to occlusion and spatial deformation. To overcome these obstacles, methods fusing RGB images with point cloud data (Mo et al., 2022; He et al., 2021; 2020; Wang et al., 2019) exhibit robustness in handling complex occlusions and textureless objects, surpassing earlier techniques in accuracy and adaptability.

Despite these advancements, existing methods still face two significant challenges. Firstly, traditional CNNs struggle to capture long-range dependencies and global context, which are crucial for accurate 6D pose estimation, as they require a deep understanding of the complex relationships between object parts. Secondly, dense fusion networks (Hua et al., 2021; He et al., 2020; Wang et al., 2019) encounter computational redundancy during multimodal data processing, stemming from their inherent intricacy in integrating diverse modal information.

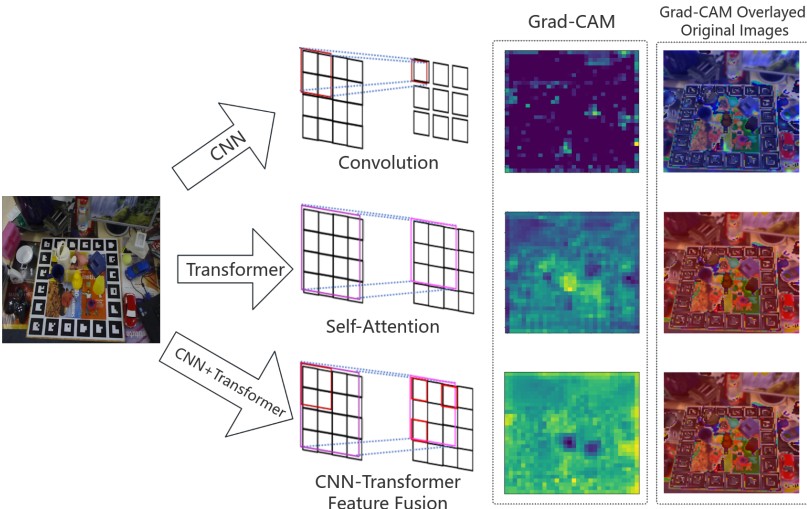

Figure 1: Comparison of CNNs, Transformers, and their combination (our method) by Grad-CAM (Selvaraju et al., 2017). The Grad-CAM images feature bright regions that highlight the areas contributing most to the network's predictions. The CNNs primarily focuses on local features, such as edges and textures, whereas the Transformers captures a wider global context. In contrast, our method achieves a more balanced extraction of both local details and global relationships.

To address these challenges, we propose two targeted approaches: 1) We explore the combination of Transformer and CNN in a hybrid architecture. Transformer excel in capturing long-range dependencies and overall context, whereas CNN specialize in extracting detailed local features. By balancing their strengths in information capture, the network achieves comprehensive feature representation and precise accuracy without the need for deep layers. 2) We design 2D convolutional networks to concurrently process RGB images and point cloud data, enabling efficient integration of local features while avoiding the complexities of dense fusion networks.

Based on these approaches, we introduce CTNet, a hybrid network designed to estimate 6D poses from RGB-D data. Its remarkable efficacy is apparent in Figure 1. We transform the depth image into an XYZ map that is aligned with the corresponding RGB image. As shown in Figure 2, the architecture of CTNet is as follows: First, we enhance the C2f (Jocher et al., 2023) and ELAN (Wang et al., 2023) modules to develop the Hierarchical Feature Extractor (HFE). Second, these local features are integrated with the XYZ map and subsequently input into a CNN-based PointNet (Qi et al., 2017a) module, which encodes the spatial information present in the point clouds. Then, a Transformer is utilized to establish global dependencies, compensating for the long-range associations often missed by CNNs. Finally, we aggregate different morphological features for pose estimation. We evaluate CTNet on LineMOD (Hinterstoisser et al., 2011) and YCB-Video (Xiang et al., 2017) datasets, which demonstrates its superiority by balancing between accuracy and inference speed. Our major contributions are as follows:

- We propose CTNet, a hybrid network that integrates CNN and Transformer architectures for 6D pose estimation, effectively capturing comprehensive feature information without the need for excessively deep network structures.

- We propose the Hierarchical Feature Extractor (HFE) for extracting local features from RGB-D data, which achieves both low computational cost and exceptional performance.

- Through comprehensive experiments, we confirm the effectiveness of our method, exhibiting strong performance on the publicly accessible LineMOD and YCB-Video datasets.

## 2 RELATED WORK

**Pose Estimation Based on RGB-D Data.** Previous methods (Hua et al., 2021; He et al., 2021; 2020; Wang et al., 2019) utilize dense fusion networks to integrate RGB and point cloud features, thereby exploiting both texture and geometry information in RGB-D data. However, dense fusion networks encounter computational redundancy during multimodal data processing, stemming from their inherent intricacy in integrating diverse modal information. Another method (Mo et al., 2022) uses 2D convolutional kernels to extract RGB and point cloud features simultaneously, similar to our approach. However, it is limited in capturing long-range dependencies due to its exclusive reliance on CNNs. In contrast, our method surpasses these limitations with a holistic strategy. We devise a CNN module for extracting local features and integrate a transformer module that captures global dependencies. By seamlessly merging local and global features, we enhance the utilization of RGB-D data, ultimately boosting the performance of pose estimation.

**Hybrid CNN-Transformer Architectures for 6D Pose Estimation.** Vision transformers, originally developed for NLP (Vaswani et al., 2017), now find widespread use in diverse computer vision tasks, such as image classification (Dosovitskiy et al., 2020), object detection (Lee et al., 2022; Liu et al., 2021; Xu et al., 2021), and semantic segmentation (Xie et al., 2021), showing considerable potential. In our approach, we deviate from methods that rely solely on visual transformers (Sandler et al., 2018). Instead, we opt to leverage transformer for capturing long-range dependencies in sequences and integrating them into the feature information for 6D pose estimation. This differs from traditional backbone networks, such as transformer-only architectures (e.g., PVT (Wang et al., 2021) and swin transformer (Liu et al., 2021)) and those entirely based on CNNs (e.g., ResNet (He et al., 2016)). Our hybrid network efficiently balances local and global feature extraction with a lightweight network, leading to a more efficient network design.

## 3 METHODS

### 3.1 NETWORK ARCHITECTURE

This paper aims to recognize rigid objects and determine their rotations $R \in SO(3)$ and translations $t \in \mathbb{R}^3$ within the camera coordinate system, utilizing a RGB-D image. To achieve this, we develop a network called CTNet, which is illustrated in Figure 2. Refer to Appendix A for input preparation and preprocessing details.

### 3.2 LOCAL FEATURE EXTRACTION

To boost efficiency and minimize parameters, we develop the Hierarchical Feature Extractor (HFE) by enhancing both the C2f (Jocher et al., 2023) and ELAN (Wang et al., 2023) architectures. This module substitutes the traditional CNN-based ResNet employed in earlier methods for local feature extraction. The network is organized into two primary segments for processing: initial feature extraction (comprising layers 1 and 2) and advanced feature extraction (comprising layers 3 and 4).

**Initial Feature Extraction.** The initial stage utilizes PConv (Chen et al., 2023) combined with an enhanced C2f module to enable efficient feature extraction, as shown in Figure 2. PConv selectively processes channels, significantly lowering memory usage and computational overhead. The channel ratio $r$ ($c_p$: $c$) determines this load. Given a feature map with dimensions $h \times w$ and a kernel size of $k$, the computational and memory access loads are minimized, as illustrated in the following equations:

$$l_1 = c_p^2 \times h \times w \times k^2, \tag{1}$$

$$l_2 = c_p \times h \times w + c_p^2 \times k^2 \approx c_p \times h \times w. \tag{2}$$

By setting $r$ to $\frac{1}{64}$, PConv reduces these loads to just 0.02% and 1.56% of that required by regular convolution, rendering them nearly negligible.

The C2f module improves feature representation by efficiently splitting, processing, and merging the feature map back together. It employs Bottleneck structures that reduce parameters while maintaining gradient flow, enabling the network to capture diverse feature scales. When combined with

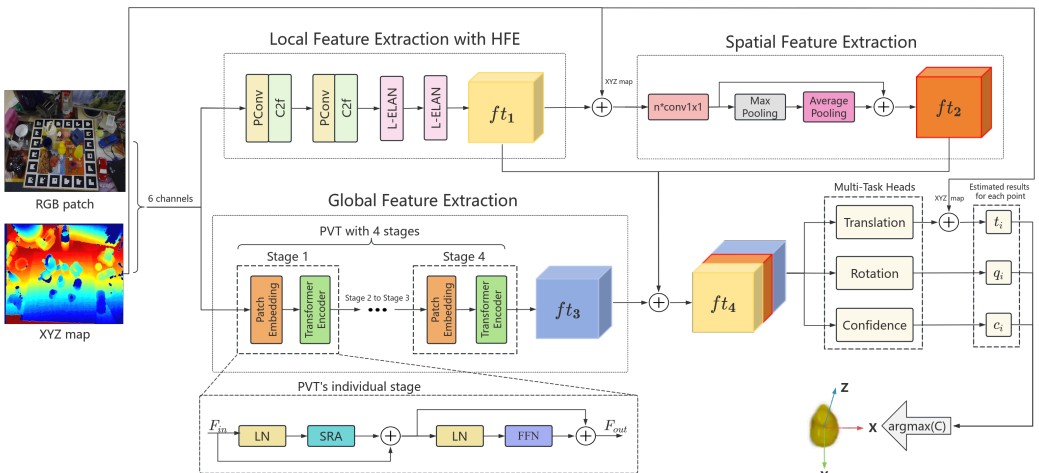

Figure 2: CTNet Overview. First, RGB-XYZ data is produced from the RGB-D image by combining color and depth information. The HFE processes the RGB-XYZ data, utilizing shallow layers for initial local feature extraction and deep layers for more complex local feature extraction. The output from the deep layers, along with the XYZ map, is fed into a PointNet-like module to extract spatial features from the point cloud. Meanwhile, the RGB-XYZ data is passed through a Pyramid Vision Transformer (PVT) to capture global features that complement those extracted by the CNNs. Finally, aggregated features are formed by amalgamating the local, spatial, and global features, and these are subsequently employed for 6D pose estimation. The pose that exhibits the maximum confidence score is chosen as the final result.

PConv, this approach yields two significant advantages: 1) richer input retention from PConv, boosting feature extraction efficiency; 2) reduced computational costs, enabling a more lightweight network without compromising accuracy.

Building on the integration of C2f and PConv, we further enhance the Bottleneck structure. We introduce two layers: a Depthwise Separable Convolution (DSC) layer and a Channel Attention Mechanism (CAM) (Woo et al., 2018) layer. Both layers incorporate normalization and activation functions to enhance feature representation. The number of Bottleneck structures is also increased to further improve feature extraction. Assuming input and output channels of $C_{in}$ and $C_{out}$, a feature map size of $h \times w$, and a kernel size of $k \times k$, the computational loads for DSC and regular convolution, denoted as $l_3$ and $l_4$, are calculated as follows:

$$l_3 = C_{in} \left( \sum_{i=1}^{n} h \times w \times k_i^2 + n \times h \times w \times C_{out} \right), \tag{3}$$

$$l_4 = h \times w \times C_{in} \times C_{out} \times k^2, \tag{4}$$

where $k_i$ is the size of the $i$-th convolutional kernel, and $n$ is the number of kernels. The computational load ratio between DSC and regular convolution is:

$$\frac{l_3}{l_4} = \frac{C_{in} \left( \sum_{i=1}^{n} h \times w \times k_i^2 + n \times h \times w \times C_{out} \right)}{h \times w \times C_{in} \times C_{out} \times k^2} = \frac{\sum_{i=1}^{n} k_i^2}{C_{out} \times k^2} + \frac{n}{k^2}. \tag{5}$$

With 64 input channels, 32 output channels, and a 3×3 kernel, the computational load of DSC is only 14.24% of that of a standard 3×3 convolution. The redesigned Bottleneck structure offers several benefits: 1) the combination of DSC and CAM provides a lightweight yet effective structure for stacking within the C2f module; 2) in shallow layers, where feature maps have high resolution, the CAM amplifies feature correlations, improving early-stage feature extraction; 3) increasing the number of Bottleneck structures enhances the network's ability to capture features across various scales, thereby improving overall feature extraction. For a detailed overview of the C2f module structure and the improved Bottleneck design, refer to Appendix B.

**Advanced Feature Extraction.** We optimize the original ELAN architecture into a lightweight variant, referred to as L-ELAN. ELAN first reduces the input data channels $F_{in}$ to $F_2$ via the CBS submodule $M_2$ (1×1 convolution). It then gathers local features using four CBS submodules ($M_3$ to $M_6$, each with 3×3 convolutions) and utilizes multiple shortcut connections to output multi-scale information $F_2$, $F_3$, and $F_4$. The ELAN module achieves strong performance by reducing parameters through channel shrinking and extracting rich features via residual connections.

In deeper layers, where channels double and features become more abstract, we design the L-ELAN to further lighten the structure. We remove the submodule $M_4$ to reduce overhead and replace the $M_2$ kernel from 1×1 to 3×3, increasing feature diversity by making $F_1$ and $F_2$ scales different. $F_3$ and $F_4$ maintain their levels with two and four 3×3 convolutions. Finally, $F_1$, $F_2$, $F_3$, and $F_4$ are concatenated and fitted by $M_7$ to produce $F_{out}$. These adjustments preserve multi-scale feature integration while reducing parameters without performance loss. Details of the ELAN and L-ELAN structures are provided in Appendix B.

### 3.3 Spatial Feature Extraction for Point Clouds

Following the extraction of local features, we utilize a CNN-based PointNet module, as referenced in (Qi et al., 2017a), to encode the spatial information inherent in the point cloud. This encoding process involves the use of $1 \times 1$ convolutions, which effectively capture both the local features and the point coordinates. To derive the spatial features of the point cloud, we subject the convolved data to max pooling and then pad it to match the size of the local features, using average pooling as our padding method.

In comparison, the method outlined in (Li et al., 2018b) relies on 2D convolutional networks to extract features from XYZ maps of point clouds. However, this approach proves less effective than heterogeneous structure methods, as documented in (Wang et al., 2019; He et al., 2020). The primary reason for this underperformance lies in the loss of spatial information that occurs during the 2D convolution operations applied to XYZ maps, as highlighted in (Mo et al., 2022).

### 3.4 Global Feature Extraction

To address the CNN network's global perception limitations, we incorporate a Pyramid Vision Transformer (PVT) (Wang et al., 2021) to capture long-range dependencies, thereby enhancing global feature extraction.

PVT employs a pyramid-like architecture, systematically reducing spatial resolution across four stages to learn multi-level features. Each stage consists of a Spatial Reduction Attention (SRA) layer and a Feed-Forward Network (FFN). The SRA utilizes a multi-head self-attention mechanism to effectively model long-distance dependencies while minimizing computational complexity by decreasing pixel count.

To process an input feature map $F \in \mathbb{R}^{H_i \times W_i \times C}$ at the $i$th stage with dimensions of height $H_i$, width $W_i$, and $C$ channels), we first perform Layer Normalization (LN) (Ba et al., 2016). The features are then rearranged into a flattened format, generating vector tokens $X \in \mathbb{R}^{N \times C}$, where $N = H_i \times W_i$, signifies the total pixel count of the feature map. The tokens $X$ are mapped to their corresponding query $Q$, key $K$, and value $V \in \mathbb{R}^{N \times C}$ vectors using trained linear mappings $W_Q$, $W_K$, and $W_V \in \mathbb{R}^{C \times C}$. To optimize memory usage, the spatial extents of $K$ and $V$ are trimmed down before applying self-attention, calculated as:

$$\text{Attention}(Q, K, V) = \text{Softmax}\left(\frac{QK^T}{\sqrt{C_{\text{head}}}}\right) V, \tag{6}$$

where $C_{head}$ denotes the channel depth per attention head in SRA. As stated in Equation 6, every token in the entire input space $F$ can interact with any other token, including itself. The global feature extraction benefits from the dual self-attention mechanism in PVT: 1) The self-attention in each transformer layer expands the network's receptive field to cover the entire image, enabling interaction between distant pixels and enhancing long-range dependency capture; 2) By embedding both depth and RGB data into each token, the self-attention mechanism evaluates pixel similarities and depth information together, allowing depth data to propagate and correct pixel errors for more accurate feature extraction.

### 3.5 6D POSE REGRESSION

The local features, spatial features of the point cloud, and global features are concatenated to form the aggregated features $F = \{f_i\}_{i=o}^N, f_i \in ^d$. Following the method in (Mo et al., 2022), we estimate rotation $R_i \in SO(3)$ and translation $t_i \in \mathbb{R}^3$ using the aggregated features $f_i$ and the corresponding visible points $\dot{p}_i \in \dot{\mathcal{P}}$. Three 1x1 convolution heads ($\mathcal{B}_\mathcal{T}$, $\mathcal{B}_\mathcal{Q}$, $\mathcal{B}_\mathcal{C}$) are employed to regress the translation offsets $\triangle \dot{t}_i \in \mathbb{R}^3$, quaternions ($q_i \in \mathbb{R}^4, \|q_i\| = 1$) and confidences $c_i \in [0, 1]$, as shown in Figure 2. Detailed formulas and processes are provided in Appendix C.

## 4 EXPERIMENTS

### 4.1 IMPLEMENTATION DETAILS

The RGB image and XYZ map are resized to 128×128. Dataset-specific training protocols are used. For LineMOD, the batch size is 8, it runs for 100 epochs, and the learning rate ranges from $5 \times 10^{-4}$ to $5 \times 10^{-6}$. As for YCB-Video, the batch size is 64, it goes through 30 epochs, and the learning rate ranges from $1.8 \times 10^{-4}$ to $1.8 \times 10^{-5}$. Details are provided in Appendix D.

### 4.2 DATASETS

**LineMOD** (Hinterstoisser et al., 2011) comprises 13 video sequences, each showcasing a unique low-textured object, serving as a benchmark for evaluating 6D object pose estimation methods (Wang et al., 2019; Vidal et al., 2018; Sundermeyer et al., 2018; Tekin et al., 2018a; Li et al., 2018c; Buch et al., 2017; Drost et al., 2010). Following segmentation practices outlined in the literature (Wang et al., 2019; Peng et al., 2019; Xiang et al., 2017), we allocate 15% of the RGB-D images for each object to the training set and reserve the remainder for testing, without incorporating any additional synthetic data.

**YCB-Video** (Xiang et al., 2017) features 21 objects captured in 92 RGB-D videos, highlighting a diverse range of object shapes and textures under various occlusion conditions. Consistent with previous studies (He et al., 2020; Wang et al., 2019; Xiang et al., 2017), we utilize 80 videos for training and select 2,949 keyframes from the remaining 12 videos as our test set, augmenting our training data with 80,000 synthetic images provided by (Xiang et al., 2017).

### 4.3 METRICS

In accordance with established practices (He et al., 2021; Wang et al., 2019; Xiang et al., 2017), we employ the ADD and ADD-S metrics for accuracy evaluation, designating ADD for non-symmetric objects and ADD-S for symmetric ones. For the LineMOD dataset, accuracy is assessed based on an ADD(S) value of less than 10% of the model's diameter, and we calculate the corresponding percentage accuracy. For the YCB-Video dataset, we utilize the area under the curve (AUC) of the ADD-S and ADD(S) metrics, varying the distance threshold from 0 cm to 10 cm to generate the accuracy-threshold curve, from which we then compute the area between this curve and the XY axes (Mo et al., 2022). Additional details regarding the ADD and ADD-S metrics are provided in Appendix E.

### 4.4 COMPARISON WITH SOTA METHODS

In this section, we comprehensively evaluate the performance of CTNet against state-of-the-art (SOTA) methods on the LineMOD and YCB-Video datasets. To provide a thorough analysis, we present both quantitative and qualitative results.

**Results on LineMOD.** The visualization results of a sample comparison between our method and ES6D on the LineMOD dataset are shown in Figure 3. The LineMOD dataset estimates the pose of objects centered on a marked board, where colored dots represent sampled points of the 3D model of the object. After pose estimation, the sampled points are projected onto the image; the closer the projected points match the target object, the more accurate the pose estimation. The highlighted areas indicate objects with significant differences between the results of the two algorithms. The

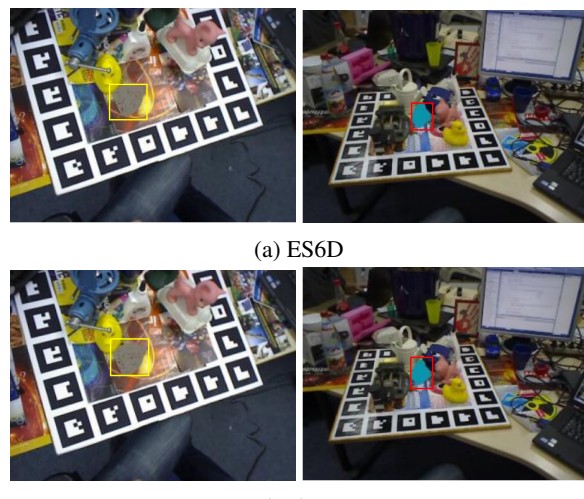

(a) ES6D

(b) Ours

Figure 3: The visualization results on LineMOD.

Table 1: Comparison of ADD(S) accuracy on LineMOD dataset.

| Object | DenseFusion | PVN3D | ES6D | Ours |
|---|---|---|---|---|
| ape | 92.3 | 95.5 | 91.4 | 95.2 |
| benchvise | 93.2 | 94.5 | 96.1 | 99.0 |
| camera | 94.4 | 94.2 | 98.0 | 99.0 |
| can | 93.1 | 94.3 | 96.0 | 100.0 |
| cat | 96.5 | 95.5 | 99.0 | 100.0 |
| driller | 87.0 | 93.3 | 97.0 | 100.0 |
| duck | 92.3 | 94.6 | 96.2 | 95.3 |
| eggbox | 99.8 | 100.0 | 99.1 | 100.0 |
| glue | 100.0 | 100.0 | 100.0 | 100.0 |
| holepuncher | 92.1 | 95.1 | 99.1 | 100.0 |
| iron | 97.0 | 92.1 | 99.0 | 97.9 |
| lamp | 95.3 | 93.7 | 99.0 | 99.0 |
| phone | 92.8 | 93.6 | 97.1 | 99.0 |
| MEAN | 94.3 | 95.1 | 97.5 | **98.8** |

visualization results demonstrate that our method generates denser and more accurate sampled points compared with ES6D.

The performance comparison of different methods on the LineMOD dataset is presented in Table 1. DenseFusion (Wang et al., 2019), PVN3D (He et al., 2020), and ES6D (Mo et al., 2022) are the current mainstream pose estimation networks. As shown, our algorithm achieves 100% or near 100% accuracy for most objects, with an average accuracy improvement of 4.5%, 3.7%, and 1.3% over DenseFusion (iterative), PVN3D, and ES6D, respectively, across the 13 objects. Notably, DenseFusion (iterative) includes an iterative refinement post-processing step, while our algorithm does not use any post-processing or refinement. This demonstrates the effectiveness of the hybrid architecture of CNN and Transformer in CTNet.

**Results on YCB-Video** The visualization results of a sample comparison between our method and ES6D on the YCB-Video dataset are depicted in Figure 4. The YCB-Video dataset estimates the pose of all objects in the scene, where colored dots represent the sampled points of each object. The highlighted areas indicate objects with significant differences between the results of the two algorithms. The visualization results demonstrate that our method generates denser and more accurate sampled points compared with ES6D.

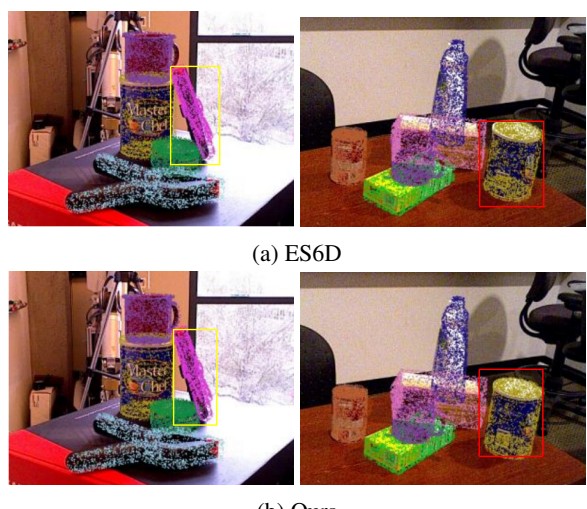

(a) ES6D

(b) Ours

Figure 4: The visualization results on YCB-Video.

Table 2: Comparison of ADD-S and ADD(S) accuracy on YCB-Video dataset.

| Object | DenseFusion | | PVN3D | | ES6D | | Ours | |
|---|---|---|---|---|---|---|---|---|
| | ADD-S | ADD(S) | ADD-S | ADD(S) | ADD-S | ADD(S) | ADD-S | ADD(S) |
| master_chef_can | 95.9 | 77.2 | 96.2 | 79.2 | 96.1 | 70.6 | 97.2 | 76.9 |
| cracker_box | 95.4 | 94.2 | 95.9 | 94.7 | 95.3 | 94.8 | 96.2 | 95.9 |
| sugar_box | 96.7 | 96.5 | 97.4 | 96.4 | 98.2 | 98.2 | 98.2 | 98.2 |
| tomato_soup_can | 97.5 | 97.4 | 96.6 | 88.5 | 95.0 | 90.2 | 96.7 | 92.4 |
| mustard_bottle | 97.9 | 94.3 | 97.4 | 96.3 | 97.9 | 97.9 | 98.2 | 97.8 |
| tuna_fish_can | 95.7 | 76.3 | 96.2 | 88.6 | 96.2 | 92.6 | 96.5 | 93.0 |
| pudding_box | 97.6 | 96.5 | 96.7 | 95.2 | 97.9 | 97.9 | 97.7 | 97.7 |
| gelatin_box | 99.0 | 97.6 | 97.8 | 96.2 | 98.6 | 98.6 | 98.9 | 98.9 |
| potted_meat_can | 90.1 | 83.7 | 93.6 | 88.3 | 92.4 | 86.0 | 93.5 | 87.0 |
| banana | 98.0 | 85.2 | 96.7 | 93.6 | 96.7 | 95.8 | 97.3 | 96.4 |
| pitcher_base | 97.1 | 96.3 | 97.1 | 96.5 | 97.6 | 97.6 | 97.3 | 97.3 |
| bleach_cleanser | 96.9 | 92.7 | 96.1 | 93.1 | 96.0 | 91.3 | 97.1 | 94.1 |
| **bowl** | 91.4 | 91.4 | 88.7 | 88.7 | 95.5 | 95.5 | 95.9 | 95.9 |
| mug | 96.2 | 91.0 | 97.5 | 95.5 | 96.5 | 94.0 | 96.7 | 94.5 |
| power_drill | 95.8 | 95.0 | 96.8 | 95.3 | 97.2 | 97.1 | 97.3 | 97.2 |
| **wood_block** | 92.6 | 92.6 | 91.5 | 91.5 | 93.7 | 93.7 | 94.1 | 94.1 |
| scissors | 86.6 | 64.4 | 96.9 | 93.5 | 90.3 | 79.2 | 90.1 | 78.2 |
| large_marker | 97.7 | 91.9 | 96.7 | 91.8 | 97.8 | 92.4 | 97.9 | 92.9 |
| **large_clamp** | 89.5 | 89.5 | 94.4 | 94.4 | 96.2 | 96.2 | 96.5 | 96.5 |
| **extra_large_clamp** | 93.3 | 93.3 | 91.1 | 91.1 | 95.2 | 95.2 | 95.2 | 95.2 |
| **foam_brick** | 92.6 | 92.6 | 96.8 | 96.8 | 95.9 | 95.9 | 97.0 | 97.0 |
| MEAN | 94.9 | 90.0 | 95.4 | 92.6 | 96.0 | 92.9 | **96.5** | **93.7** |

Table 2 displays the ADD-S and ADD(S) AUC values, as well as their averages, for 21 objects (with symmetrical objects highlighted in bold) in the YCB-Video dataset across various methods. The experimental outcomes demonstrate that our algorithm outperforms DenseFusion, PVN3D, and ES6D by an average accuracy improvement of 3.7%, 1.1%, and 0.8% on ADD(S), and by 1.6%, 1.1%, and 0.5% on ADD-S, respectively. Given the challenging occluded scenarios in the YCB-Video dataset, these results indicate that CTNet exhibits strong robustness in handling such cases. The accuracy results for both datasets are presented as line graphs in Appendix F.

### 4.5 ABLATION STUDIES

We further analyze the contribution of individual modules in CTNet by comparing six different network configurations, as detailed in Table 3. The results show that the complete architecture,

Table 3: Ablation studies of CTNet. IFEL: initial feature extraction layers, including PConv and C2f modules; AFEL: advanced feature extraction layers, featuring the L-ELAN module; SIE: spatial information encoding; PVT: pyramid vision transformer.

| Method | IFEL | | AFEL | SIE | PVT | LineMOD ADD(S) | YCB ADD(S) | Time (ms) | FLOPs (G) | Parameters (M) |
|---|---|---|---|---|---|---|---|---|---|---|
| | PConv | C2f | L-ELAN | | | | | | | |
| Unified like (Li et al., 2018a) | | | | | | 96.0 | 91.5 | 20.3 | 7.39 | 17.85 |
| CTNet_1 | | ✓ | | | | 97.5 | 92.9 | 15.7 | 6.5 | 14.9 |
| CTNet_2 | ✓ | ✓ | | | | 97.6 | 93.0 | 14.3 | 6.3 | 14.9 |
| CTNet_3 | ✓ | ✓ | ✓ | | | 97.9 | 93.1 | 11.3 | 2.3 | 5.7 |
| CTNet_4 | ✓ | ✓ | ✓ | ✓ | | 98.3 | 93.4 | **11.1** | **2.7** | **6.1** |
| CTNet_5 | ✓ | ✓ | ✓ | ✓ | ✓ | **98.8** | **93.7** | 12.5 | 3.6 | 6.4 |

CTNet_5, which integrates IFEL, AFEL, SIE, and PVT, achieves the best overall performance. In contrast, the Unified like (Li et al., 2018a) underperforms in both accuracy and inference speed. Notably, CTNet_2, which introduces PConv layers, demonstrates a substantial improvement over CTNet_1, highlighting the positive interaction between PConv and C2f. Furthermore, CTNet_3, which adds the AFEL, significantly reduces FLOPs while maintaining accuracy, highlighting the efficiency of the L-ELAN design. Although the inclusion of SIE in CTNet_4 and PVT in CTNet_5 slightly increases FLOPs compared to CTNet_3, the gains in accuracy outweigh these additions. This confirms the complementary nature of local, spatial, and global feature extraction, reinforcing the robustness of CTNet's hybrid architecture.

Table 4: Practical effects of applying the HFE to other 6D pose estimation frames.

| Method | ADD-S | ADD(S) | Time(ms) | FLOPs(G) | Parameters(M) |
|---|---|---|---|---|---|
| DenseFusion (origin) | 94.8 | 90.1 | 39.9 | 11.8 | 17.2 |
| DenseFusion (HFE) | 96.4 | 93.1 | 14.1 | 2.7 | 5.4 |
| PVN3D (origin) | 95.4 | 92.6 | 199.6 | 190.8 | 31.1 |
| PVN3D (HFE) | 96.5 | 92.8 | 82.6 | 90.5 | 9.0 |
| ES6D (origin) | 97.5 | 92.9 | 15.8 | 6.7 | 15.1 |
| ES6D (HFE) | 98.5 | 93.2 | 12.6 | 3.7 | 7.0 |

To showcase the adaptability of HFE within CTNet, we substituted the CNN components of three prominent algorithms with our novel HFE and carried out experiments on the YCB-Video dataset. The outcomes, detailed in Table 4, reveal that incorporating HFE boosts inference speed by 64.7%, 58.6%, and 20.3% for DenseFusion, PVN3D, and ES6D, respectively. Furthermore, FLOPs are decreased by 77.1%, 52.6%, and 44.8%, and parameter counts are reduced by 68.6%, 71.1%, and 53.6% for these respective frameworks. Notably, accuracy also improves across all three systems. These findings highlight HFE's exceptional transferability and efficiency, establishing it as a versatile and effective component for 6D pose estimation networks.

# 5 CONCLUSIONS

This paper introduces CTNet, a hybrid architecture combining CNN and Transformer components for 6D pose estimation. The CNN-based Hierarchical Feature Extractor (HFE) optimizes local feature extraction, and the Pyramid Vision Transformer (PVT) captures a broader global context. Along with CNN-based PointNet spatial encoding, this design achieves superior pose estimation performance. Experiments on the LineMOD and YCB-Video datasets show that CTNet balances accuracy and efficiency, surpassing current models. Furthermore, HFE enhances other architectures when integrated, proving its robustness and adaptability.

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

## A    INPUT PREPARATION AND PREPROCESSING

Following the method in (Mo et al., 2022), we use the segmentation network from PoseCNN (Xiang et al., 2017) to obtain masks and bounding boxes for target objects. Each mask and RGB-D image patch, cropped by the bounding boxes, is used as input. TThe masked depth pixels are normalized, converted into an XYZ map, and concatenated with the RGB patch to form a 6-channel input.

## B    DETAILED STRUCTURES OF KEY MODULES

In this section, we introduce the specific designs of C2f, its Bottleneck structures, as well as ELAN and L-ELAN modules.

**C2f Module**    The C2f module (Jocher et al., 2023) is a component for extracting features at multiple levels. As depicted in Figure 5, the initial phase involves passing the feature map through a CBS submodule, which integrates Conv, BatchNorm, and SiLU elements. This submodule contains a $1 \times 1$ convolutional kernel, a normalization layer, and an activation layer. Following this, the module incorporates the CSP design, dividing the feature map into two paths. One path is directly forwarded to the Concat module, while the other undergoes Bottleneck processing before merging with the first path in the Concat module.

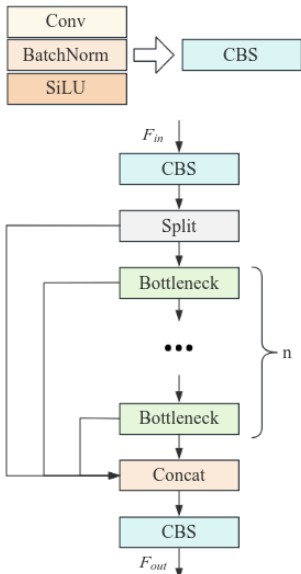

Figure 5: The framework of C2f module.

**Bottleneck Structure within C2f**    The Bottleneck structure is composed of two convolutional layers linked by a residual connection. As shown in Figure 6, this structure initially decreases the channel count and subsequently reinstates it. In our implementation, we replace the traditional convolutional layers in the compression and expansion phases with depthwise separable convolutional layers and channel attention mechanism layers, respectively.

**ELAN and L-ELAN Modules**    The differences between ELAN (Wang et al., 2023) and L-ELAN are clearly visible in Figure 7. Specifically, in L-ELAN, we replace the $1 \times 1$ convolutional kernel in component $M_3$ with $3 \times 3$ kernel, enhancing its ability to capture spatial context. Additionally, we remove component $M_4$ entirely, simplifying the architecture and improving computational efficiency. These modifications enhance the practicality of the L-ELAN module.

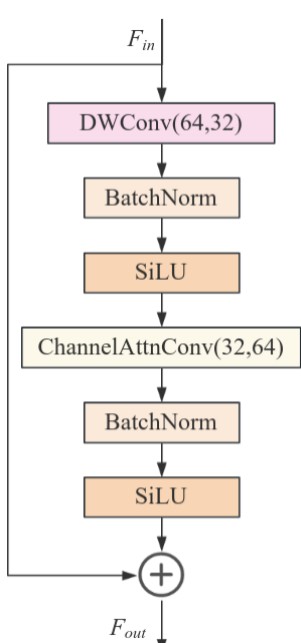

Figure 6: Improved Bottleneck structure in the C2f module.

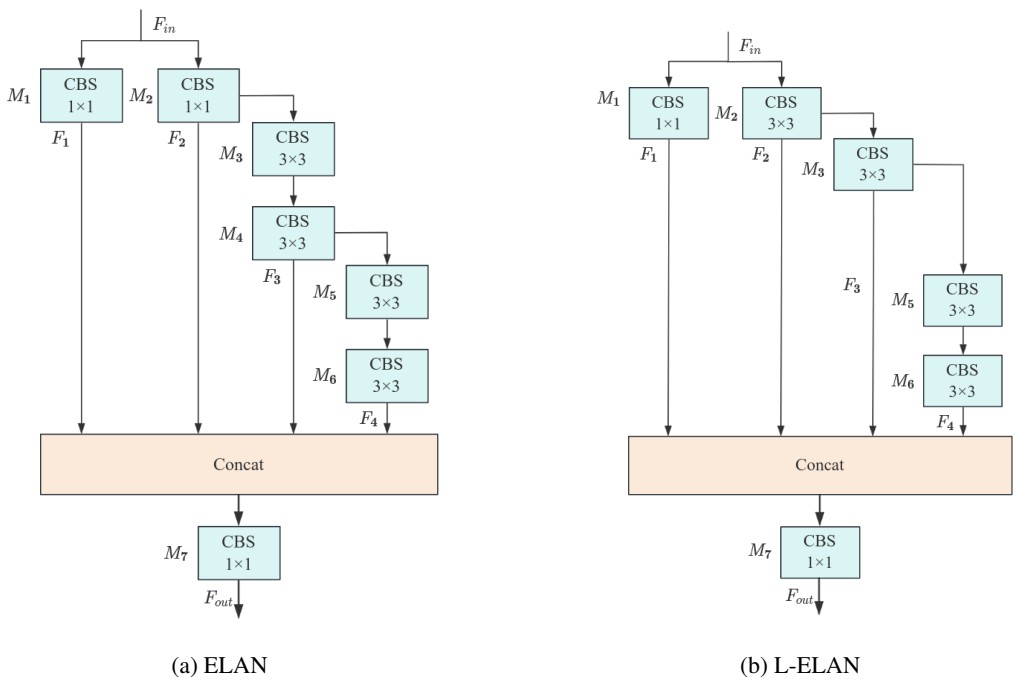

(a) ELAN

(b) L-ELAN

Figure 7: The frameworks of ELAN and L-ELAN module.

## C  6D POSE REGRESSION

Following the method in (Mo et al., 2022), we estimate rotation $R_i \in SO(3)$ and translation $t_i \in \mathbb{R}^3$ using the aggregated features $f_i$ and the corresponding visible points $\dot{p}_i \in \dot{\mathcal{P}}$. Three 1x1 convolution heads ($\mathcal{B}_\mathcal{T}$, $\mathcal{B}_\mathcal{Q}$, $\mathcal{B}_\mathcal{C}$) are used to regress the translation offsets $\triangle t_i \in \mathbb{R}^3$, quaternions ($q_i \in \mathbb{R}^4, \|q_i\| = 1$) and confidences $c_i \in [0, 1]$, as shown in Figure 2.

**3D translation regression.** Treating the origin of the normalized object coordinate system as a virtual keypoint, the translation $t_i$ can be obtained by calculating the offset $\triangle \dot{t}_i$ between the visible points $\dot{p}_i$ and the origin. This equation can be expressed as:

$$\dot{t}_i = \mathcal{B}_{\mathcal{T}}(f_i), \tag{7}$$

$$t_i = \frac{(\dot{p}_i + \triangle \dot{t}_i)}{\gamma} + p_c, \tag{8}$$

here the offset distribution of the visible points $\dot{p}_i$ is within a specific sphere.

**3D rotation regression.** We use quaternions as the rotation representation following (Xiang et al., 2017; Wang et al., 2019). The rotation matrix we obtain is as follows:

$$R_i = Quaternion\_matrix(Norm(\mathcal{B}_{Q\}}(f_i)), \tag{9}$$

$$Norm(q_i) = \frac{q_i}{\|q_i\|}, \tag{10}$$

where $Quaternion\_matrix(\cdot)$ represents the function that converts quaternions into a rotation matrix (Sarabandi & Thomas, 2019).

**Confidence regression.** To determine the optimal regression results, we set up a confidence estimation head to evaluate the confidence $c_i$ of each feature. The equation is as follows:

$$c_i = Sigmoid(\mathcal{B}_C(f_i)), \tag{11}$$

where we used the self-supervised method mentioned in (Wang et al., 2019) to train the confidence branch $\mathcal{B}_C$.

## D    IMPLEMENTATION DETAIL

The RGB image and XYZ map are uniformly resized to 128×128. The training protocol varies by dataset. For the LineMOD dataset, we use a batch size of 8 and train the model for 100 epochs, starting with an initial learning rate of $5 \times 10^{-4}$, which decays to $5 \times 10^{-6}$ using a cosine annealing schedule from the 90th epoch onward. A linear warm-up is applied during the first epoch, gradually increasing the learning rate from $5 \times 10^{-6}$ to $5 \times 10^{-4}$. For the YCB-Video dataset, we set the batch size to 64 and train the model for 30 epochs, beginning with an initial learning rate of $1.8 \times 10^{-4}$, which decays using a cosine annealing schedule and is maintained at $1.8 \times 10^{-5}$ from the 20th epoch until the end of training. A linear warm-up phase increases the learning rate from $1.8 \times 10^{-6}$ to $1.8 \times 10^{-4}$ during the first epoch. These training strategies ensure efficient convergence and optimal performance of the model across different datasets.

## E    METRICS

We follow the evaluation methods used in (Xiang et al., 2017; Wang et al., 2019; He et al., 2021), employing the average distance metrics ADD and ADD-S to assess the accuracy of the algorithm. The ADD metric is calculated by computing the average distance between the transformed object vertices using the predicted pose $[R, T]$ and the ground truth pose $[R^*, T^*]$:

$$ADD = \frac{1}{m} \sum_{x \in O} \|(Rx + T) - (R^*x + T^*)\|, \tag{12}$$

where $x$ is a point in the object point cloud $O$ and $m$ is the number of points in the point cloud. However, the ADD metric can only be applied to non-symmetric objects with unique true values. For symmetric objects, which have multiple equivalent true poses, we use the ADD-S, which is invariant to symmetry. The ADD-S calculation is as follows:

$$ADD - S = \frac{1}{m} \sum_{x_1 \in O} \min_{x_2 \in O} \|(Rx_1 + T) - (R^*x_2 + T^*)\|. \tag{13}$$

To comprehensively evaluate the performance of our algorithm, we consider an ADD(S) less than 10% of the model diameter as the criterion for a correct estimation on the LineMOD dataset and calculate the percentage accuracy. For the YCB-Video dataset, we employ the AUC of the ADD-S and ADD(S) metrics, adjusting the distance threshold from 0 cm to 10 cm to generate the accuracy-threshold curve and subsequently calculate the area between this curve and the XY axes (Mo et al., 2022).

# F ACCURACY RESULTS ON BOTH DATASETS

Figure 8 compares the performance of CTNet with three other advanced methods on the LineMOD and YCB-Video datasets. In this comparison, the X-axis represents the object categories in the datasets, along with the average performance across all categories (MEAN). The Y-axis signifies the ADD(s) metric, which quantifies the accuracy of object recognition. Evidently, the red line, which represents CTNet, demonstrates superior recognition accuracy for most objects in comparison to the other three methods. This notable difference underscores the precision advantage that our approach offers.

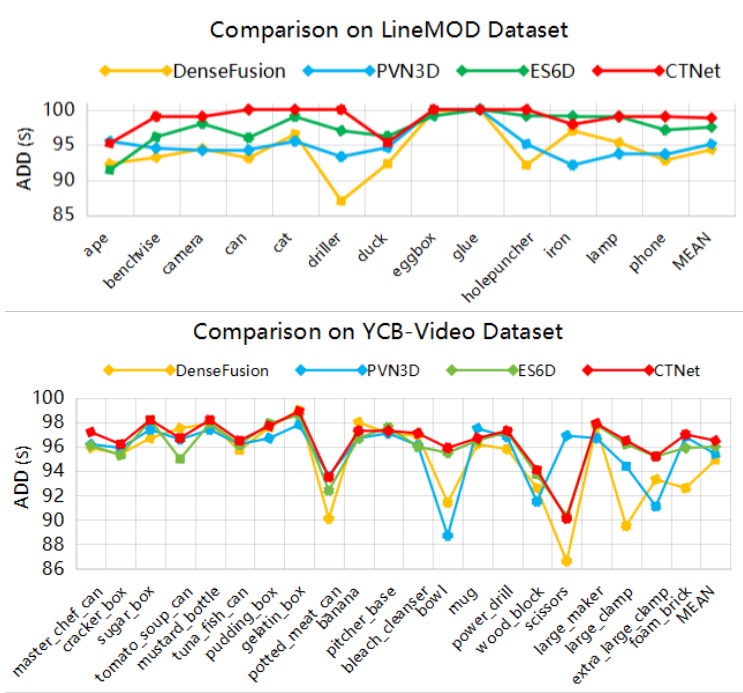

Figure 8: Performance comparison on LineMOD and YCB-Video datasets.

