# OpenReview forum: "CTNet: A CNN-Transformer Hybrid Network for 6D Object Pose Estimation"
_ICLR.cc/2025/Conference — Submitted to ICLR 2025_

### Official Review · Reviewer_J91x · 2024-10-22

**Soundness:** 1
**Presentation:** 2
**Contribution:** 1
**Rating:** 1
**Confidence:** 5

**Summary:**

This paper introduces a hybrid architecture combining CNN and Transformer for estimating object poses from a single RGBD image. A Hierarchical Feature Extractor (HFE) is proposed for extracting local features with low computational cost. The authors show that this method could achieve reasonable results on some common object pose estimation datasets.

**Strengths:**

++ The work done in this paper could further verify that combining the strenghs from both CNN and Transformer is beneficial for object pose estimation.

**Weaknesses:**

-- In the abstract, the meanings of C2f and ELAN are not clearly defined, which may lead to confusion for readers.

-- Some works for using CNN+Transformer for object pose estimation are missing in the discussion of related work, such as [a], [b] and [c]. As the idea of combinating CNN and Transformer architectures for object pose estimation is not new, the first contribution of the work (L99-101) is not an actual contribution.


-- Some works for RGB-D based pose estimation are missing, such as [d], [e], [f], and [g].


-- The experiments are insufficient. Currently, the metrics on the Linemod dataset are nearly saturated and only an additional YCB-Video dataset for the comparison is insufficient.

-- It seems the numbers for the compared methods reported in this paper could not match those in previous works. The comparison to the state-of-the-art might be unfair.

-- The methods included for comparison are a little bit outdated (it seems no works from 2023/2024 are included). For example, methods like [d] and [g] should be considered for comparison.

-- According to [h], metrics like ADD or ADD(-S) have certain drawbacks. Recent works such as [d] tend to use the more comprehensive BOP metrics for evaluation. Please also use the BOP metrics for evaluation in this work.

-- The whole method has a very incremental novelty. The main novelty lies in the network architecture, however, it is a combination of many existing modules with minor improvements. The performance on the datasets is not state-of-the-art, which also confirms that the proposed architecture offers limited improvements over existing solutions.

-- The improvements in this paper do not seem to be specifically designed for the task of object pose estimation. The authors could further evaluate the effectiveness of the proposed method on other tasks taking RGBD images as input, such as RGBD-based object detection and segmentation.

-- For the metric names on YCB-Video, the AUC should not be omitted, for example, in Table 2 and Table 3.

[a] Sun et al., FGCT6D: Frequency-Guided CNN-Transformer Fusion Network for Metal Parts’ Robust 6D Pose Estimation. RA-L 2024.

[b] Fu et al., Hybrid6D: A Dual-Stream Transformer-CNN Approach for 6D  Object Pose Estimation from RGB-D Images. ROBIO 2023.

[c] Zhang et al., LaPose: Laplacian Mixture Shape Modeling for RGB-Based Category-Level Object Pose Estimation. ECCV 2024.

[d] Lin et al., HiPose: Hierarchical Binary Surface Encoding and Correspondence Pruning for RGB-D 6DoF Object Pose Estimation. CVPR 2024.

[e] Shen et al., HFE-Net: hierarchical feature extraction and coordinate conversion of point cloud for object 6D pose estimation. Neural Computing and Applications 2024.

[f] Liu et al., RaSim: A Range-aware High-fidelity RGB-D Data Simulation Pipeline for Real-world Applications. ICRA 2024.

[g] Zhou et al., Deep Fusion Transformer Network with Weighted Vector-Wise Keypoints Voting for Robust 6D Object Pose Estimation. ICCV 2023.

[h] Hodan et al., BOP Challenge 2020 on 6D Object Localization. ECCVW 2020.

**Questions:**

-- The input RGBD images are resized intro 128x128. Are they cropped object regions or the whole images? Are there any detection or segmentation methods required? How does the method tackle the problem of estimating the poses for multiple objects in one image?

**Details Of Ethics Concerns:**

The metric numbers of existing approaches included in this paper could not match those in the original papers. This might result in an unfair comparison.

---

### Official Review · Reviewer_VwnJ · 2024-10-29

**Soundness:** 2
**Presentation:** 3
**Contribution:** 2
**Rating:** 3
**Confidence:** 4

**Summary:**

This paper introduces a hybrid network, called CTNet, which leverages the strengths of both CNNs and Transformers for 6D object pose estimation. In CTNet, the CNN branch captures local features, enhanced by a hierarchical feature extractor, while the Transformer branch captures long-range context. Additionally, a PointNet module is incorporated to provide spatial information. Experiments on the LineMOD and YCB-V datasets demonstrate the effectiveness of CTNet.

**Strengths:**

- CTNet combines the strengths of CNNs and Transformers to enhance feature extraction, enabling more precise pose estimation.
- The paper is clearly organized.

**Weaknesses:**

- Limited novelty: The combination of CNNs and Transformers has been widely explored across various fields in computer vision, like [1], [2],  [3]. However, there is insufficient discussion comparing CTNet with existing methods, including a lack of experimental comparisons, particularly in the related work and experiments sections.

- Suboptimal performance: Some state-of-the-art methods, such as FFB6D [4], DFTr [5], HiPose [6], and RDPN [7], are not included in Tables 1 and 2, several of which achieve higher accuracy. Additionally, comparisons with results on the Occlusion-LineMOD dataset should be included in the paper.

- Limited ablation studies: Comparisons on speed and model size with existing pose estimation methods, particularly those using only CNNs or Transformers, are needed to highlight the advantages of the proposed hybrid network. Additionally, the improvements shown in Table 3 are modest, particularly on the YCB-V dataset.


[1] Fang et al., "A Hybrid Network of CNN and Transformer for Lightweight Image
Super-Resolution".

[2] Chen et al., "LEFormer: A Hybrid CNN-Transformer Architecture for Accurate Lake Extraction from Remote Sensing Imagery".

[3] Zhang et al., "Lite-Mono: A Lightweight CNN and Transformer Architecture for
Self-Supervised Monocular Depth Estimation".

[4] He et al., "FFB6D: A Full Flow Bidirectional Fusion Network for 6D Pose Estimation".

[5] Zhou et al., "Deep Fusion Transformer Network with Weighted Vector-Wise
Keypoints Voting for Robust 6D Object Pose Estimation".

[6] Lin et al., "HiPose: Hierarchical Binary Surface Encoding and Correspondence Pruning
for RGB-D 6DoF Object Pose Estimation".

[7] Hong et al., "RDPN6D: Residual-based Dense Point-wise Network for 6Dof Object Pose
Estimation Based on RGB-D Images".

**Questions:**

- The meanings of 'l_1', 'l_2', 'c_p' and 'c' are not explained in Eq (1) and (2).

- It's not clear how to use average pooling for padding in Sec. 3.3.

---

### Official Review · Reviewer_cXXV · 2024-11-01

**Soundness:** 3
**Presentation:** 2
**Contribution:** 2
**Rating:** 3
**Confidence:** 5

**Summary:**

This paper focuses on the 6D pose estimation with low computational complexity. The key observation is that the existing methods struggle to capture long-range dependencies and have the intricacy of integrating diverse modal information. This paper proposes that CTNet combine the CNN and Transformer to extract hybrid backbone features for pose estimation. The experimental results demonstrate that the proposed CTNet outperforms the baseline with lower computational complexity.

**Strengths:**

1. The proposed CTNet as an advanced backbone successfully improves the performance of 6D pose estimation with lower computational complexity.
2. The ablation study is well-designed to validate the effectiveness of the proposed method.

**Weaknesses:**

1. Limited Knowledge Contribution to Pose Estimation Field: The primary contribution of this work appears to rely heavily on the universality of an improved backbone rather than on advancing pose estimation-specific knowledge. While leveraging a stronger backbone has been shown in prior studies to enhance model performance across tasks, this work does not provide a substantial methodological or conceptual contribution specifically tailored to the pose estimation domain.
2. Insufficient Technical Novelty: The method proposed in this paper integrates several high-performing modules from existing backbones without introducing substantial innovations in either architecture design or task-specific adaptations. Merely combining known techniques, albeit effectively, does not present a strong enough level of technical novelty or innovation, limiting its impact.
3. Lack of Comprehensive Comparisons: To substantiate the effectiveness of the proposed backbone for pose estimation, it would be essential to conduct comparative experiments against a more extensive range of recent backbones or similar architectures. Including a broader set of relevant baselines would allow for a more accurate assessment of the contributions and competitive advantages of the proposed method.
4. Missing important related work discussion and comparison. The paper lacks a thorough discussion and comparison with other CNN-transformer hybrid networks specifically designed for pose estimation, such as DFTr [1]. Additionally, it does not include comparisons with state-of-the-art (SOTA) methods in the pose estimation field [1, 2].
[1] Zhou, Jun, et al. "Deep fusion transformer network with weighted vector-wise keypoints voting for robust 6d object pose estimation." Proceedings of the IEEE/CVF International Conference on Computer Vision. 2023.
[2] Hong, Zong-Wei, Yen-Yang Hung, and Chu-Song Chen. "RDPN6D: Residual-based Dense Point-wise Network for 6Dof Object Pose Estimation Based on RGB-D Images." Proceedings of the IEEE/CVF Conference on Computer Vision and Pattern Recognition. 2024.
5. The visualization in Figures 3 and 4 does not have significant differences.  Why does the method have denser sampled points?

**Questions:**

See Weakness

[Update]
Due to unresolved concerns raised in my initial review and additional critical issues highlighted by other reviewers, I have revised my score downward.

---

### Official Review · Reviewer_vt8f · 2024-11-08

**Soundness:** 1
**Presentation:** 1
**Contribution:** 2
**Rating:** 3
**Confidence:** 3

**Summary:**

This paper considers the task of 6DOF object detection from RGBD images with CNN+Transformer architectures. The main contribution is the development of the architecture, with various design choices that optimize for efficient compute. Important design choices include the use of pyramidal transformers and partial convolutions (that leave some channels unprocessed).

**Strengths:**

The strongest aspect of this paper is the diagnostic evaluation in Table 3, which reveals the impact of various design choices. Compared to  prior art that serves as a baseline implementation, the design choices improve accuracy by ~2 percent (from 96->98 and from 91->93) while reducing latency by 2X and the parameter count by 2.5X.

I consider the 2 percent delta significant when viewed as a reduction of error (e.g., 2X reduction in error).

Results are presented on flagship datasets for 6DOF estimation, including YCB-Video and LineMod.

**Weaknesses:**

Many of the design choices are not self-contained, and refer to other architectures (C2F and ELAN) that are not particularly well-known. Overall, I found the description of the design quite hard to follow. This is my major complaint about the paper. For example, the CBS submodule is mentioned in Line 218 without any background, but requires going to the appendix for any explanation.

The paper is missing references to well-known and state-of-the-art work in this space such as FoundationPose [CVPR 2025]. As a point of reference, I found that paper to far easier to follow, both in terms of the technical contribution and overall narrative. Its empirical performance is similar to the presented work (outperforming on one metric but underperforming on another).

This paper clearly wants to position itself as an architecture paper, but the writing is so convoluted that it makes it hard to articulate the key architectural novelty. The idea of building transformers on top of CNN features has been widely explored in the community (SWIN transformers, Vision Transformers for Dense Prediction), and it seems like the design choices explored here serve mainly to reduce the number of parameters and latency by 2X. I do not view such reductions as sufficient for publication (thought I admit the error reduction is significant).

**Questions:**

I would like the authors to clearly articulate the technical novelty of the proposed hierarhical feature extractor "HFE" module that differs from past work.

---

### Meta-Review · Area_Chair_4SKK · 2024-12-23

**Metareview:**

The paper proposes a hybrid CNN and Transformer architecture to improve 6D object pose estimation with lower computational complexity. While the paper does achieve those goals, it relies on well-known developments in prior works and misses out on positioning with respect to important prior works with very similar aims and methods.

**Additional Comments On Reviewer Discussion:**

All reviewers point out the above concerns to recommend rejection and no author rebuttal is submitted. The AC agrees with the reviewer consensus that the paper may not be accepted at ICLR.

---

### Decision · Program_Chairs · 2025-01-22

Reject